# The Development of a Regulator of Human Serine Racemase for N-Methyl-D-aspartate Function

**DOI:** 10.3390/biomedicines12040853

**Published:** 2024-04-12

**Authors:** Lu-Ping Lu, Wei-Hua Chang, Yi-Wen Mao, Min-Chi Cheng, Xiao-Yi Zhuang, Chi-Sheng Kuo, Yi-An Lai, Tsai-Miao Shih, Teh-Ying Chou, Guochuan Emil Tsai

**Affiliations:** 1Department of Research and Development, SyneuRx International (Taiwan) Corp., New Taipei 221416, Taiwan; luping.lu.ls10@nycu.edu.tw (L.-P.L.); weihua.chang@syneurx.com (W.-H.C.); ellie.mao@syneurx.com (Y.-W.M.); mickey.cheng@syneurx.com (M.-C.C.); xiaoyi.zhuang@syneurx.com (X.-Y.Z.); archie.kuo@syneurx.com (C.-S.K.); yalai1027@gmail.com (Y.-A.L.); cindy.shih@syneurx.com (T.-M.S.); 2Institute of Biochemistry and Molecular Biology, National Yang Ming Chiao Tung University, Taipei 112304, Taiwan; 3Graduate Institute of Clinical Medicine, Taipei Medical University, Taipei 11031, Taiwan; 4Department of Pathology and Precision Medicine Research Center, Taipei Medical University Hospital, Taipei Medical University, Taipei 112304, Taiwan; 5Department of Psychiatry and Biobehavioral Science, UCLA School of Medicine, Los Angeles, CA 90024, USA

**Keywords:** serine racemase, racemization, β-elimination, NMDA, enzyme activator, tannic acid, malonate

## Abstract

It is crucial to regulate N-methyl-D-aspartate (NMDA) function bivalently depending on the central nervous system (CNS) conditions. CNS disorders with NMDA hyperfunction are involved in the pathogenesis of neurotoxic and/or neurodegenerative disorders with elevated D-serine, one of the NMDA receptor co-agonists. On the contrary, NMDA-enhancing agents have been demonstrated to improve psychotic symptoms and cognition in CNS disorders with NMDA hypofunction. Serine racemase (SR), the enzyme regulating both D- and L-serine levels through both racemization (catalysis from L-serine to D-serine) and β-elimination (degradation of both D- and L-serine), emerges as a promising target for bidirectional regulation of NMDA function. In this study, we explored using dimethyl malonate (DMM), a pro-drug of the SR inhibitor malonate, to modulate NMDA activity in C57BL/6J male mice via intravenous administration. Unexpectedly, 400 mg/kg DMM significantly elevated, rather than decreased (as a racemization inhibitor), D-serine levels in the cerebral cortex and plasma. This outcome prompted us to investigate the regulatory effects of dodecagalloyl-α-D-xylose (α12G), a synthesized tannic acid analog, on SR activity. Our findings showed that α12G enhanced the racemization activity of human SR by about 8-fold. The simulated and fluorescent assay of binding affinity suggested a noncooperative binding close to the catalytic residues, Lys56 and Ser84. Moreover, α12G treatment can improve behaviors associated with major CNS disorders with NMDA hypofunction including hyperactivity, prepulse inhibition deficit, and memory impairment in animal models of positive symptoms and cognitive impairment of psychosis. In sum, our findings suggested α12G is a potential therapeutic for treating CNS disorders with NMDA hypofunction.

## 1. Introduction

### 1.1. Role of NMDA Receptor and Its Regulatory Enzyme Serine Racemase

Serine racemase (SR) is an enzyme that regulates D-serine, a co-agonist of the N-methyl-D-aspartate (NMDA) receptor, in the central nervous system (CNS). SR catalyzes not only D-serine synthesis by the racemization of L-serine but also D-serine degradation by β-elimination of D-serine. L-serine can be degraded by β-elimination too. An intriguing dual-base mechanism has been proposed for SR, whereby Lys56 serves as the si-face base, alpha-deprotonating an appropriately oriented external aldimine of L-serine, giving rise to a common, cofactor-stabilized quinonoid intermediate. Subsequent re-protonation by the putative re-face base, Ser84, leads to the D-serine racemization product, whereas expulsion of the (presumably protonated) beta-OH leaving group leads to pyruvate, the β-elimination product [1,2].

The NMDA receptor plays an important role in brain development including long-term potentiation, synaptic plasticity, learning, and memory formation [3,4]. NMDA receptor activation requires the occupation of both the glutamate and “glycine” binding sites. D-serine, in addition to glycine, is the potent endogenous agonist for the “glycine” binding site. In fact, the co-agonist site should be named “D-serine and/or glycine binding site”. The physiological relevance of glycine vs. D-serine likely depends on both microscopic and macroscopic anatomy. D-serine and SR have parallel anatomical distribution to NMDA receptors, enriched in corticolimbic regions, while glycine is ubiquitously distributed to the whole brain including the brain stem and spinal cord. Glycine is located close to the extrasynaptic NMDA receptor, while D-serine is located close to the synaptic NMDA receptor. The regulation of NMDA function also depends on the preferential affinity of synaptic NMDA receptors for D-serine and extrasynaptic NMDA receptors for glycine [5,6,7].

### 1.2. NMDA Receptor Dysfunction and Its Potential Drug Target

It has been postulated that D-serine is relevant to neurotoxicity while glycine is more for physiological processes like long-term potentiation. Regulating D-serine levels through SR can be a novel drug target for the treatment of neurotoxic disorders like stroke and traumatic brain injury or neurodegenerative disorders including amyotrophic lateral sclerosis (ALS) and late-stage Alzheimer’s disease (AD) with elevated D-serine [8,9,10,11,12,13,14]. At the same time, treatment with an NMDA receptor agonist, D-serine alone, or D-serine combined with a glycine reuptake inhibitor has been demonstrated to improve the psychotic symptoms and cognition in patients with schizophrenia, depression, early-stage AD, and aging-associated cognitive decline [7,15,16,17,18]. In a double-blind crossover clinical trial, D-serine treatment could alleviate symptoms of schizophrenia and depression [19]. However, high-dose administration or long-term treatment with D-serine may cause nephrotoxicity and hepatotoxicity leading to proteinuria and transaminitis [20,21,22].

SR, as a metabolic regulator of D-serine, is an unexplored frontier for investigating the modulation of NMDA function. It can serve as a target for the development of novel drugs in the field where the treatment of CNS disorders with NMDA dysfunction is unmet. Upregulating the racemization of SR can raise D-serine levels and benefit CNS diseases associated with low NMDA function. Consistently, SR-knockout mice have minimal D-serine in CNS and reveal synaptic disarrangement and cognitive and developmental deficits [23,24]. Above and beyond, we aim to modulate the D-serine levels in CNS bivalently by regulating the unique dual mechanism of SR, racemization vs. β-elimination, and provide a novel therapeutic target that is critical for CNS disorders with NMDA dysregulation.

### 1.3. Research Objectives

In this study, we treated C57BL/6J male mice intravenously with dimethyl malonate (DMM), a pro-drug of the most known competitive SR inhibitor malonate [25,26,27,28]. We observed that the IV administration of DMM, a pro-drug of malonate, is supposed to inhibit SR in vitro [27,29], in fact, elevated D-serine levels in the cerebral cortex and plasma of C57BL/6J mice. This finding prompted us to synthesize the SR inhibitor dodecagalloyl-α-D-xylose (α12G) and investigate its regulatory effect on SR activity. Our results revealed that α12G increased the racemization of human serine racemase (hSR) by about 8-fold. Simulated and fluorescent assays of binding affinity suggested a noncooperative binding close to the hSR catalytic residues Lys56 and Ser84. Moreover, α12G treatment can improve CNS disorders with NMDA hypofunction including hyperactivity, prepulse inhibition deficit, and memory impairment in animal models of positive symptoms and cognitive impairment. Thus, these findings suggested α12G as a potential therapeutic agent for CNS disorders with NMDA hypofunction. Importantly, the regulation of SR needs to take into account racemization and β-elimination for both D- and L-serine.

## 2. Materials and Methods

### 2.1. Analysis of D-Amino Acid (DAA) Concentration in C57BL/6J Mice Brain

For the determination of DAA concentration, C57BL/6J male mice received 50, 100, 200, and 400 mg/kg DMM (TGI, Tokyo, Japan) IV, and the mice were euthanized 30 min later. Cerebral cortex and plasma samples (*n* = 3 − 9) were collected and stored at −80 °C until the time of analysis. The brain tissues were weighed and homogenized on ice with 5-fold volumes of 1000 ng/mL tolbutamide (internal standard, IS, Sigma-Aldrich, St. Louis, MO, USA)) in methanol (Merck KGaA, Darmstadt, Germany). The plasma was combined with 4-fold volumes of 1250 ng/mL IS in methanol and vortexed for 2 min. The homogenates were centrifuged at 12,000× *g* rpm for 10 min at 4 °C in an Eppendorf 5417R centrifuge (Eppendorf, Hamburg, Germany). The supernatant was passed through a 0.22 μm filter (Merck KGaA, Darmstadt, Germany) and analyzed by liquid chromatography–tandem mass spectrometry (LC-MS/MS) using the following method: An Agilent 1260 LC quaternary pump with an Agilent 1260 LC infinity autosampler and an Agilent 1100 column oven (Marshall Scientific, Hampton, NH, USA) were used to inject the samples onto an Astec Chirobiotic column (5 μm, 250 × 4.6 mm I.D., Sigma-Aldrich, St. Louis, MO, USA). The mobile phase was a 20/80 mixture of mobile phase A (0.1% formic acid (Sigma-Aldrich, St. Louis, MO, USA) in ddH_2_O) and mobile phase B (0.1% formic acid in acetonitrile), and the run was isocratic at a flow rate of 0.3 mL/min. Quantitation was achieved by MS/MS detection in positive ion multiple reaction monitoring (MRM) mode for analyte and IS using an AB SCIEX 3200 triple quadrupole ion trap mass spectrometer (AB Sciex, Framingham, MA, USA). The parameters for curtain gas, ion spray voltage, temperature, ion source gas 1, and ion source gas 2 were 10 psi, 5000 V, 550 °C, 35 psi, and 30 psi, respectively. AB SCIEX Analyst*^®^* (version 1.6.3, AB Sciex, Framingham, MA, USA) software was used for system control and data processing.

### 2.2. Purification of Recombinant hSR

Recombinant wild-type hSR was constructed with a 6-histidine tag at the 3′ terminus. In detail, the original construct was subcloned into pET42b using the 5′BamHI and 3′NotI restriction sites. The first PCR amplified hSR by introducing 5′NdeI and 3′XhoI sites and skipping the stop codon. The PCR amplicon was then subcloned into pET42b cleaved by NdeI and XhoI restriction sites, which removed all tags at the 5′ terminus of pET42b. The second PCR step amplified the whole plasmid using a set of 5′phosphorylation primers that skipped 6 nucleotides of the XhoI site and 2 histidine tags out of 8 histidine tags of pET42b to generate the correct sequence.

BL21 cells containing pET42b-hSR were grown in Luria-Bertani (LB) medium (CHUMEIA, Hsinchu County, Taiwan) at 37 °C until OD_600_ reached 0.6. The culture flask was then placed on ice and induced with 0.5 mM Isopropyl β-D-1-thiogalactopyranoside (IPTG) (Cyrusbioscience, Taipei, Taiwan) for protein expression at 25 °C for 20 h. The cells were harvested by centrifugation, and the pellet was resuspended in lysis buffer (20 mM Tris-HCl, 100 mM NaCl, 20 mM imidazole, 0.1 mM PMSF, pH 8.0). Cells were disrupted by a high-pressure cell homogenizer NanoLyzer N2 (Gogene Corp., Hsinchu County, Taiwan). The crude protein was centrifuged at 4 °C and 8000× *g* for 30 min. hSR was further purified from soluble extract followed by an NTA fast flow HisTrap column (Cytiva, Marlborough, MA, USA), a HiTrap Q column (Cytiva, Marlborough, MA, USA), and a HiLoad 16/60 Superdex 200 prep grade column (Cytiva, Marlborough, MA, USA). The purified hSR in wash buffer (20 mM Tris-HCl, pH 8.0, 100 mM NaCl, 2 mM MgCl_2,_ 50 µM pyridoxal 5′-phosphate (PLP), and 5 mM Dithiothreitol (DTT)) was concentrated to ~11 mg/mL and stored at −80 °C.

#### 2.2.1. SDS-PAGE and Native Polyacrylamide Gel Electrophoresis

Recombinant hSR (1 μg) was mixed with sample buffer (2×) containing 62.5 mM Tris-HCl (pH 6.8), 25% glycerol, and 1% bromophenol blue. All samples were subjected to SDS-PAGE and native polyacrylamide gel which did not contain sodium dodecyl sulfate at 80 volts and 4 °C for 100 min. The gel was then stained with Coomassie R-250 Brilliant Blue dye and destained with deionized water until the background of the gel was clear.

#### 2.2.2. Preparation of α12G

The synthetic routes of the α12G are illustrated in Figure 1. A detailed description of synthesis is in the Appendix A. The intermediate **5** was prepared in a four-step reaction starting from gallic acid. Gallic acid was esterified under sulfuric acid/methanol to afford methyl 3,4,5-trihydroxybenzoate (**1**). The adjacent two phenol groups were protected with α,α-dichlorodiphenylmethane to obtain intermediate **2**. The remaining free phenol group was then reacted with allyl bromide to give the fully protected monogalloyl motif **3**. The methyl ester was saponified to produce compound **4**. The carboxylic compound was reacted with oxalyl chloride to give acyl chloride **5**, which was submitted to the next step within a short time. The digallic acid **11** was prepared in a six-step process starting from compound **5**. Compound **5** was reacted with tert-butoxide to provide the tert-butyl ester **6**. The allyl group was then deprotected with tetrakis(triphenylphosphine)palladium and aniline to afford compound **7**. A coupling reaction between **7** and **4** under mild, non-acidic Steglich conditions gave the digalloyl motif **8**. The compound was then submitted to palladium-catalyzed cleavage of allyl ether, giving compound **9**. The free phenol group was protected with benzyl bromide to provide compound **10**. The removal of the tert-butyl ester protecting group in the presence of a high concentration of formic acid afforded fully protected digallic acid **11**. To synthesize tetragalloyl-α-D-xylose **12**, α-D-(+)-xylose was acylated with acyl chloride **5** using pyridine as the base. After removing the allyl protecting groups from the four galloyl moieties, the installation of intermediate **11** was performed under Steglich conditions, providing the fully protected dodecagalloyl-α-D-xylose **14**. Finally, the simultaneous removal of both diphenylmethylene ketals and benzyl groups under hydrogenolytic conditions in tetrahydrofuran at room temperature yielded the desired dodecagalloyl-α-D-xylose (α12G).

### 2.3. Two-Step Enzymatic Assay

This assay was designed by the formation of D-serine by hSR and the oxidation of D-serine by D-amino acid oxidase (DAAO) and horseradish peroxidase (HRP) with the peroxidase substrate Amplex Red^TM^ (10-acetyl-3,7-dihydroxyphenoxazine) (Thermo Fisher Scientific, Lagoas Park, Porto Salvo, Portugal). The hSR activity assay reaction was carried out in 100 mM HEPES buffer, pH 8.0, containing 2.4 µg of hSR protein, 50 µM PLP, 1 mM MgCl_2_, 0.25 mM ATP, 400 mM L-serine, and 2 times serially diluted α12G (8 mM to 0.2 µM). Compound-free wells were used as blank controls in the assay. All reactions were performed in duplicates. The reaction mixtures were incubated at 37 °C for 4 h. The reaction was stopped by heating the reaction mixtures at 95 °C for 10 min and storage at 4 °C. A reaction mixture for the second-step enzymatic assay, DAAO assay, with a final volume of 100 µL was made with a 10 µL sample from hSR assay, 45 µM FAD, 10 units of HRP, 60 ng DAAO, 10 mM lead acetate, 10 mM Amplex Red^TM^, and 100 mM Tris HCl (pH = 8.5). Lead acetate was used to conjugate α12G as the latter would inhibit the DAAO function in the fluorometric assay, leading to an underestimate of the effects of α12G on hSR. The fluorescence was detected by the TECAN Infinite 200 Microplate Reader (GMI, MN, USA). The racemization percentage was calculated as follows: racemization % = (F_sample, 20 min_ − F_sample, 0 min_)/(F_blank, 20 min_ − F_blank, 0 min_) × 100%.

### 2.4. DAAO Enzymatic Assay

A reaction mixture for the DAAO assay, with a final volume of 100 µL was made with 1 µL 2 times serially diluted α12G (4 µM to 7.8 nM), 45 µM FAD, 10 units of HRP, 60 ng DAAO, 10 mM Amplex Red^TM^, 5 mM D-serine, and 100 mM Tris HCl (pH = 8.5). All reactions were performed in duplicates. The fluorescence was detected by a fluorometric meter (TECAN Infinite 200). The racemization percentage was calculated as follows: activity % = (F_sample, 20 min_ − F_sample, 0 min_)/(F_blank, 20 min_ − F_blank, 0 min_) × 100%.

### 2.5. Molecular Modeling

The crystal structure of hSR was modified from the complex with PDB ID 5X2L, which contains open-form hSR in the complex with its coenzyme, PLP. The coenzyme PLP and water molecules were removed using Chimera 1.17.1 [30] prior to the docking run. Dock 6.10 was utilized for ligand−protein complex simulations. The docked molecule α12G is (2*R*,3*R*,4*S*,5*R*)-tetrahydro-2*H*-pyran-2,3,4,5-tetrayl tetrakis(3-((3,4-dihydroxy-5-((3,4,5-trihydroxybenzoyl)oxy)benzoyl)ox)-4,5-dihydroxybenzoate) that was synthesized and mentioned above. The stereo conformations of α12G were energy minimized by Avogadro (Avogadro: an open-source molecular builder and visualization tool. Version 1.2 http://avogadro.cc/) using the MMFF94 force field and algorithms with default settings, and the grid generation was conducted following the Rizzo Lab tutorial (2021 DOCK tutorial 1 with PDBID 1HW9) with its default settings. Top-ranking binding modes and Grid scores were selected and visualized using UCSF Chimera version 1.17.1.

### 2.6. Intrinsic Tryptophan Fluorescence (ITF)

First, 30 µL of 2 µM hSR protein was added into 270 µL buffer (100 mM Tris-HCl, pH 8.0, 1 mM MgCl_2_, 0.25 mM ATP, 50 µM PLP, 10% DMSO) with ligand and subsequently reacted for 20 min at room temperature. All samples were measured in triplicates and excited at 282 nm. The emission spectra were measured with a 5 nm bandwidth resolution at 300 to 400 nm wavelengths by a Varioskan™ LUX multimode microplate reader (Thermo Fisher Scientific, Lagoas Park, Porto Salvo, Portugal). The fluorescence intensity collected at 304 nm was analyzed using GraphPad Prism 8 software (Dotmatics, Boston, MA, USA). The dissociation constant (K_d_) was derived by specific binding with the Hill slope model: Y = B_max_ × X^n^/(Kd^n^ + X^n^), with X being the ligand concentration, maximum number of binding sites (B_max_) being the maximum specific binding, and *n* being Hill slope.

### 2.7. Animals

C57BL/6J male mice were housed below 5 mice per cage and maintained on a 12/12 h light/dark cycle at 20–26 °C and 30–70% humidity. All behavioral tests were performed during the dark cycle. All animals used were 8–12 weeks old. The study protocol of all behavioral tests was approved (approval number: SR-111-003, approval date: 24 August 2022) by the Institutional Animal Care and Use Committee (IACUC) of SyneuRx Animal Study Center (Organization number 290, New Taipei City, Taiwan).

### 2.8. Drug Administration

For behavioral studies, α12G was dissolved in 35% PEG-400 and administrated by oral gavage 20 min prior to the MK-801 insult. MK-801 (Sigma-Aldrich, St. Louis, MO, USA) was administrated via intraperitoneal (IP) injection 20 min prior to the behavioral tests, with the dosage of 0.2 mg/kg for the open field test and novel object recognition test and 0.3 mg/kg for the prepulse inhibition test.

### 2.9. Open Field Test

C57BL/6J male mice were randomly assigned into six groups, 7–12 mice per group, (1) saline control, (2) MK-801, (3) 1 mg/kg α12G + MK-801, (4) 3 mg/kg α12G + MK-801, (5) 10 mg/kg α12G + MK-801, and (6) 30 mg/kg α12G + MK-801. The mice were placed in a Plexiglas cage (37.5 × 21.5 × 18 cm), and their spontaneous locomotor activities were measured for 60 min using a Photobeam Activity System (PAS) open field (San Diego Instruments, San Diego, CA, USA). The total photo beam breaks of each mouse were measured as an index of locomotor activity.

### 2.10. Prepulse Inhibition Test

C57BL/6J male mice were randomly assigned into five groups, 6–21 mice per group, (1) saline control, (2) MK-801, (3) 1 mg/kg α12G + MK-801, (4) 3.5 mg/kg α12G + MK-801, and (5) 10 mg/kg α12G + MK-801. Prepulse inhibition (PPI), a measurement of sensorimotor gating, served as a biomarker of schizophrenia. Using the SR-LAB startle apparatus (San Diego Instruments, San Diego, CA, USA), the PPI ratio with startle response was elicited by various acoustic stimuli. Each C57BL/6J male mouse was placed in an acrylic cylinder in a dark and sound-insulated chamber. With a constant 65 dB background noise, the session comprised a 5 min acclimation period followed by 4 blocks including a total of 64 trials. The pulse alone (PA) trial consisted of a 40 msec acoustic pulse at 120 dB; a 20 msec non-startling prepulse (pp) at 71 dB (pp6), 75 dB (pp10), or 83 dB (pp18) preceding a 120 dB burst with 100 msec interval was presented in the pp-P trials; the non-stimulus (NS) trials presented no stimulus. Both the first and the final block contained six PA trials, while the other two blocks were composed of PA, pp-P, and NS trials in pseudo-random order at an averaged 15 msec interval (varying from 10 to 20 s). The ratio of PPI was calculated from the startle response by the following formula: PPI % = 100 × [(PA score) − (pp-P score)]/(PA score), while the PA score was averaged from PA values in the two middle blocks.

### 2.11. Novel Object Recognition

C57BL/6J male mice were randomly assigned into five groups, 6–8 mice per group, (1) saline control, (2) MK-801, (3) 1 mg/kg α12G + MK-801, (4) 3 mg/kg α12G + MK-801, and (5) 10 mg/kg α12G + MK-801. A novel object recognition test (NOR) was conducted in a square chamber (40 × 40 × 40 cm) under 15–20 lux light intensity, and a digital camera was used to record the behaviors. The test consisted of two sessions. In session I, mice were allowed to explore the chamber with two identical objects (yellow square Lego bricks) for 5 min and were then moved to the home cage for 5 min. In session II, mice were allowed to explore the chamber with one object kept the same (familiar object) and the other replaced with a novel object (blue circular bottle cap) for 10 min. The interaction of mice with both the objects (familiar vs. novel) was recorded, and the discrimination index was calculated with the following formula:Discrimination index = Time (novel object) − Time (familiar object)/Total exploring time

## 3. Results

### 3.1. In Vivo Regulation of hSR by DMM

To investigate the regulation of racemization by hSR, we applied DMM, the pro-drug of malonate, to C57BL/6J male mice. The mice were euthanized 30 min after the administration. The cerebral cortex and plasma were harvested and analyzed by LC-MS/MS to quantify D-serine and L-serine levels after the treatment with DMM (Figure 1). It was found that 400 mg/kg DMM significantly raised D-serine levels in the cerebral cortex 1.43 times (*p* < 0.001; *n* = 3; Student’s *t*-test) as well as in plasma 1.75 times (*p* < 0.05; *n* = 3; Student’s *t*-test), presumably by promoting the racemization activity of SR. L-serine levels were also raised in the cerebral cortex 1.44 times (*p* < 0.001; *n* = 3; Student’s *t*-test) and in plasma 1.55 times (*p* < 0.05; *n* = 3; Student’s *t*-test).

### 3.2. Purification of the Recombinant hSR Protein

Recombinant wild-type hSR protein was constructed with a 6-Histidine tag at the 3′ terminus. The expressed protein was purified on a nickel-charged affinity resin. The expression of the protein yielded about 2–3 mg of purified protein from 1 L bacteria culture. To further characterize the dimer structure of hSR, the purified protein was subjected to ion exchange and size-exclusion chromatography. Upon gel-filtration chromatography on a Superdex 200 column (Cytiva, Marlborough, MA, USA), the purified SR preparation gave a clear symmetric peak corresponding to a theoretical mass of ~74 kDa, likely corresponding to the SR dimer (Figure 2). This strategy of combining three chromatography methods yielded hSR protein that was ~99% pure. We used SDS-PAGE and native gel electrophoresis to confirm the molecular weight and dimer structure of hSR, respectively. The result showed purified hSR as a single band corresponding to the hSR dimer structure (Figure 2).

### 3.3. Enzymatic Characterization of hSR

The activity of hSR was determined by the velocity of L-serine racemization to D-serine. D-serine was then assayed by a fluorometric assay. The enzyme catalytic reaction in which hSR participates has three different reactions including L-serine dehydration, L-serine racemization, and D-serine dehydration. Previous experiments showed that SR activity was triggered by ATP, which is the modulator of hSR catalysis and regulates the enzymatic activity. The catalytic efficiency of L-serine dehydration is dramatically enhanced by the addition of 2 mM ATP in the presence of 200 mM L-serine at 37 °C, which leads to a 31-fold increase in enzyme activation, with a change in k_cat_/K_M_ from 8.1 to 253 s^−1^ M^−1^. However, the enhancements of D-serine dehydration and L-serine racemization increase only 4-fold (0.6 to 2.4 s^−1^ M^−1^) and 1.9-fold (9.2 to 17.5 s^−1^ M^−1^), respectively, in the presence of 2 mM ATP. Therefore, we aimed to find out the optimal ATP concentration that gives rise to the highest racemization. Our result showed that within the physiological range of ATP concentration, 0.25 mM ATP promotes the most racemization by hSR (Figure 3A). Under such an ATP concentration, the enzymatic activities lean toward racemization.

### 3.4. In Vitro Regulatory Function of α12G on hSR

We attempted to study the effects of α12G on hSR activity using the optimal 0.25 mM ATP concentration, which promotes racemization most effectively, to determine the half-maximal effective concentration (EC_50_) of α12G.

The two-step enzymatic assay described in Section 2.3 was designed by the formation of D-serine by hSR and the consequent oxidation of D-serine by DAAO. The inhibition of DAAO, which also degrades D-serine, by α12G will lead to overestimating the effects of racemization by hSR. We determined the inhibitory effects of α12G on DAAO at IC_50_ = 0.16 µM (Figure 3B). Therefore, lead acetate was applied to conjugate α12G for the determination of racemization. The concentration-dependent response curve for the racemization reaction (% on the Y axis, drug concentration on the X axis) exhibits an inverted U-shape (Figure 3C). α12G promoted hSR racemization under the concentration of 2 mM. The highest racemization activity was at the concentration of 31.25 µM, which was defined as EC_100_ (Figure 3C). The two EC_50_ values of α12G on hSR racemization are 8.82 and 220.80 µM (both concentrations reached EC_50_). On the contrary, when the concentration of α12G is higher than 2 mM, the racemization activity of hSR is inhibited.

### 3.5. Interactions between the Active Site Residues of hSR with α12G

In the in silico simulations, hSR was selected for targeting with a focused pocket on its substrate binding site. α12G was applied for molecular modeling using DOCK 6.10 and an open-form hSR crystal structure (PDB: 5X2L). The rotatable bonds of α12G were set as fixed. The chemical structure of α12G and its simulated binding affinity are shown in Figure 4. The distances and binding peptides of five predicted hydrogen bonds are as follows: Asn86 (2.400 Å), Lys114 (2.647 Å), Pro233 (2.694 Å), Lys241 (2.019 Å), and Lys241 (2.639 Å). The Grid score for the binding mode is −164.83 kcal/mol. These findings predicted strong affinities of α12G with hSR.

### 3.6. The Binding Affinity of α12G for hSR

Under a 0.25 mM ATP concentration, which promotes racemization of hSR the most, we found that α12G has a binding affinity (K_d_) of 0.24 µM and a B_max_ = 96.86 fmol/mg for hSR. The Hill coefficient (*n*) of 1.28 is close to *n* = 1, indicating a noncooperative binding. This suggested that the ligand molecule is probably prone to bind to a single binding site with no cooperativity.

### 3.7. Pharmacodynamic Effects of α12G on Rodent NMDA Hypofunction Models

To investigate the neurobehavioral effects of α12G in vivo, MK-801, a non-competitive NMDA receptor antagonist, was applied to introduce hypofunction of NMDA in C57BL/6J mice. Several behavioral tests corresponding to the positive symptoms and cognitive impairment of schizophrenia were conducted.

First, an open field test, which measures locomotor activity to simulate psychomotor agitation, a typical positive symptom, was performed in the rodent model with 0.2 mg/kg MK-801 administration. α12G alleviated the MK-801-induced hyperactivity (*p* < 0.001; *n* = 8; Dunnett’s multiple comparisons test) (Figure 5A). Secondly, disruption of PPI, a biomarker for cognitive impairment in CNS disorders with NMDA hypofunction, was induced in the rodent model with 0.3 mg/kg MK-801 administration to investigate the potency of α12G. At the dose of 1 mg/kg, α12G can significantly alleviate PPI deficits at PPI 6 and PPI 10 (*p* < 0.05; *n* = 14; Student’s *t*-test) (Figure 5B). Thirdly, in the novel object recognition test, the MK-801-treated group showed no preference between novel and familiar objects. In addition, 10 mg/kg α12G treatment attenuated this MK-801-induced memory impairment (*p* < 0.01; *n* = 8; Student’s *t*-test) (Figure 5C).

## 4. Discussion

It is crucial to regulate NMDA function bivalently depending on the CNS conditions. NMDA hyperfunction is involved in the pathogenesis of neurotoxic disorders such as stroke, traumatic brain injury, or neurodegenerative disorders with elevated D-serine levels like ALS and late-stage AD. On the contrary, NMDA-enhancing agents such as D-serine itself and glycine reuptake inhibitors have been demonstrated to improve psychotic symptoms and cognition in CNS disorders with NMDA hypofunction like schizophrenia, depression, aging-associated cognitive decline, and early-stage AD. However, since a high dose of D-serine may induce nephrotoxicity and hepatotoxicity, an alternative therapeutic approach might serve as a drug design strategy.

hSR emerges as a promising target for the bidirectional regulation of NMDA function. It can produce D-serine by converting L-serine through racemization. In addition, it can catabolize both D-serine and L-serine into pyruvate by β-elimination. In this study, we developed a molecular regulator of hSR confirmed by molecular docking, enzymatic assay, and neurobehavioral studies.

We first observed that the administration of DMM, a pro-drug of malonate, supposed to inhibit SR in vitro [27,29], in fact, elevated D-serine levels in both the cerebral cortex and plasma of C57BL/6J mice. The result prompted us to further investigate the mechanism of racemization activation. We synthesized α12G as a potential therapeutic agent for CNS disorders with NMDA hypofunction. α12G is a structural analog of tannic acid which is also a DAAO inhibitor that can improve NMDA function in CNS disorders [31]. The inhibitory effects of α12G on DAAO were tested with an IC_50_ of 0.16 µM (Figure 3B). We tested the regulatory effects of α12G on hSR by a commonly used two-step enzymatic assay, which involves sequential incubation with hSR and then DAAO [29,32,33,34]. α12G enhances the racemization of hSR about 8-fold in the two steps of enzymatic assay. The EC_100_ is 31.25 µM and the EC_50_ is 8.82 and 220.80 µM of α12G on hSR’s multiple reactions. Intriguingly, at concentrations exceeding 2 mM, α12G exhibited inhibitory effects on the racemization activity of hSR (Figure 3C).

Our findings reveal α12G to be the first reported compound that can serve as both an activator and inhibitor of SR (Figure 3C). Enzyme activators are defined as chemical compounds that increase the velocity of enzymatic reactions while the catalysts increase reaction rates without altering the chemical equilibrium. The actions of activators are the opposite of the effect of inhibitors [35]. According to the binding affinity from the post-docking and ITF experiments, the binding site of α12G on hSR is likely located in the catalytic pocket, leading to the physical barriers that keep away its substrate (Figure 4A). Our result shows the Hill coefficient (*n*) of 1.28 which is close to *n* = 1, suggesting a noncooperative binding (Figure 4B). The results declare that α12G probably binds to a single binding site with no cooperativity and is close to the catalytic residues, Lys56 and Ser84.

The reason α12G mediates both D-serine production and degradation is based on the unique dual mechanism of SR racemization and β-elimination (dehydration). The racemization converts L-serine to D-serine, and the β-elimination degrades D-serine (and/or L-serine) to pyruvate. The catalytic efficiency of dehydration of L-serine is higher with a k_cat_/K_M_ of 253 s^−1^ M^−1^ than D-serine dehydration (k_cat_/K_M_ = 2.4 s^−1^ M^−1^) and L-serine racemization (k_cat_/K_M_ = 17.5 s^−1^ M^−1^), in the presence of 2 mM ATP. The ratio of L-serine β-elimination and racemization efficiency is about 14 [32]. It indicates that SR, highly expressed in the CNS, tends to regulate the physiological D-serine level by controlling the availability of its racemization substrate, L-serine, by β-elimination rather than facilitating the production of D-serine by racemization. This is consistent with the hypothesis that D-serine is essential for NMDA function, but excess D-serine can be toxic, while D-serine is an obligatory co-agonist that needs to be rigorously regulated.

At the same time, we hypothesize that the effect of DMM and α12G on enhancing D-serine levels results from the combination of relatively more inhibition of L-serine β-elimination and limited influence toward L-serine racemization to D-serine. This is supported by the catalytic efficiency order of K_cat_/K_M_ of L-serine β-elimination, L-serine racemization, and D-serine β-elimination [32]. The inverted U-shape dose–effect relationships in Figure 3C can be dissected into three parts. First, the racemization activity increases from the basal level to the maximum, indicating relatively more inhibition of L-serine β-elimination, which leads to an overall accumulation of L-serine and, consequently, more D-serine production. Second, this activation of racemization peaks and continues to decline, showing a dose-dependent increase in subsequent racemization inhibition that counteracts the effects of L-serine accumulation. Third, the transition from a racemization-activated effect to an inhibitory effect reflects an indiscriminating inhibition of β-elimination and racemization when the concentration of α12G reaches 2 mM. This is due to the overall increase in the inhibitory effect and even leads to an indiscriminating inhibition of β-elimination and racemization when α12G exceeds 2 mM.

The findings of the two-step enzymatic assay revealed the increased hSR racemization activity is consistent with our in vivo neurochemical findings. This hypothesis of the racemization increase is also supported by the observed elevation of both L-serine and D-serine in the cerebral cortex and plasma of mice administrated with DMM (Figure 1B). The accumulation of L-serine can facilitate the dynamic equilibrium of racemization and favor the production of D-serine. Taken together, these findings illuminate the inverted U-shape dose–effect relationships that were also observed in the learning and memory behavioral tests when regulation of SR was involved [36,37].

hSR is a brain-region-specific enzyme that exhibits high expression in the forebrain regions of corticolimbic circuitry. On the contrary, DAAO is highly expressed in the hindbrain regions and minimally expressed in the forebrain regions [38]. This might explain why the basal level of D-serine is quite high in the forebrain regions, including the cortex. The relatively lower expression of DAAO likely favors a higher concentration of D-serine in forebrain regions.

Our findings support that elevating D-serine levels through regulating hSR is a promising approach to enhancing NMDA function. Though several SR inhibitors and activators have been confirmed for regulating racemization [29,33,34], in vivo evaluation is scarce. Understanding the true effects of these compounds on D-serine regulation requires considering their potential inhibition of DAAO in the commonly used two-step enzymatic assay, as this could lead to an over- or underestimation of their effects on SR activity. We investigated the compound x-0458 (2-morpholinoaniline), which was reported to inhibit SR at an IC_50_ of 10.6 µM in the two-step enzymatic assay [33]. However, we found that it also has an inhibitory effect on DAAO. To address this confounding factor, we applied lead acetate to conjugate α12G and avoided the concomitant DAAO inhibitory effect on the assay.

In conclusion, we identified α12G ((2*R*,3*R*,4*S*,5*R*)-tetrahydro-2*H*-pyran-2,3,4,5-tetrayl tetrakis(3-((3,4-dihydroxy-5-((3,4,5-trihydroxybenzoyl)oxy)benzoyl)ox)-4,5-dihydroxybenzoate)) as a potential hSR regulator evaluated by in silico, in vitro*,* and in vivo assays. In general, a therapeutic approach to CNS disorders with NMDA hypofunction has been a rare find. Our study provides compelling evidence supporting the benefits of NMDA enhancement by the regulation of SR; α12G treatment can improve NMDA hypofunction in animal models of positive symptoms and cognitive impairment (Figure 5).

To validate hSR as a therapeutic target and develop α12G as a novel therapeutic, several limitations need to be addressed. First, the molecular mechanism by which α12G enhances the racemization reaction and its exact binding on the enzyme’s structure needed to be further elucidated. An X-ray crystallography study of the hSR-α12G complex in combination with site-directed mutagenesis will be pursued to further discover the molecular mechanism of α12G’s regulatory effect on hSR. Second, the inverted U-shaped relationship between the concentration of hSR’s regulators and their activator and/or inhibitory function needed to be confirmed and further explored. The dose–response assessment of α12G on several NMDA dysfunction animal models does indicate (part of) the inverted U-shaped dose–response. However, this dose–response needed to be further confirmed. Third, long-term studies are necessary to evaluate the potential chronic effects and toxicity besides the single administration of α12G. Fourth, an analysis of α12G with other known SR modulators can be pursued to compare α12G’s efficacy and safety profile with those of compounds targeting the same pathway. Fifth, electrophysiological studies measuring long-term potentiation (LTP) or long-term depression (LTD) can be explored to prove the downstream effects of α12G on synaptic plasticity, learning, and memory. Sixth, the behavioral assessments in this study are limited. NMDA dysregulation is common in CNS disorders; the inclusion of other disease models or behavioral tasks is needed to provide a more comprehensive understanding of α12G’s therapeutic potential. Last, pharmacokinetic studies comparing different administration routes can be pursued to increase the clinical relevance and potential for translational application.

## Data Availability

The data presented in this study are available on request from the corresponding author due to commercial restrictions.

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
