# Peer review of "The Development of a Regulator of Human Serine Racemase for N-Methyl-D-aspartate Function"

_biomedicines, 2024, doi:10.3390/biomedicines12040853_

Round 1

Reviewer 1 Report

Comments and Suggestions for Authors

1. The study provides substantial evidence on the potential of α12G as a therapeutic agent for CNS hypofunction disorders through the regulation of serine racemase activity. However, the mechanism by which α12G enhances the racemization reaction and its effects on the enzyme's structure could be further elucidated. Specifically, additional experiments, such as X-ray crystallography or cryo-EM studies of the serine racemase-α12G complex, could provide more precise insights into the structural basis of α12G's modulatory effects.

2. The inverted U-shaped dose-response curve is intriguing and suggests a complex mechanism of action. It would be beneficial to explore a broader range of concentrations, especially in the transition regions where α12G shifts from being an activator to an inhibitor. This could help in refining the therapeutic dose range and understanding the concentration-dependent dynamics more accurately.

3. While the acute pharmacodynamic effects of α12G are promising, long-term studies are necessary to evaluate the potential chronic effects and toxicity. Chronic administration studies in rodent models, assessing both behavioral outcomes and potential nephrotoxicity or hepatotoxicity, would be valuable.

4. The paper would benefit from a comparative analysis of α12G with other known serine racemase modulators, both activators and inhibitors. This comparison could help contextualize α12G's efficacy and safety profile relative to other compounds targeting the same pathway.

5. Additional experiments to investigate the downstream effects of α12G on synaptic plasticity, learning, and memory would strengthen the paper. Electrophysiological studies measuring long-term potentiation (LTP) or long-term depression (LTD) in hippocampal slices from treated animals could provide direct evidence of cognitive enhancement.

6. The use of open field and prepulse inhibition tests provides initial evidence of α12G's effects on NMDA receptor-mediated behaviors. Expanding the behavioral assessment to include tests for anxiety-like behavior, depression-like symptoms, and other cognitive tasks such as maze learning would provide a more comprehensive profile of α12G's therapeutic potential.

7. The study focuses on intravenous administration of α12G. Exploring alternative administration routes, such as oral or intranasal, could increase the clinical relevance and potential for translational application of the findings. Pharmacokinetic studies comparing different administration routes would be an excellent addition to the paper.

Overall, this study represents a significant step forward in the development of modulators for serine racemase as therapeutic agents for CNS disorders. Addressing these suggestions could further solidify the findings and pave the way for translational and clinical research in this promising area.

Author Response

  1. The study provides substantial evidence on the potential of α12G as a therapeutic agent for CNS hypofunction disorders through the regulation of serine racemase activity. However, the mechanism by which α12G enhances the racemization reaction and its effects on the enzyme's structure could be further elucidated. Specifically, additional experiments, such as X-ray crystallography or cryo-EM studies of the serine racemase-α12G complex, could provide more precise insights into the structural basis of α12G's modulatory effects.

A: Thank you for your insightful comment. We do plan to operate an X-ray crystallography experiment to further discover the molecular mechanism of α12G’s regulatory function on hSR. We add this point in the discussion in the last part of the discussion.

  1. The inverted U-shaped dose-response curve is intriguing and suggests a complex mechanism of action. It would be beneficial to explore a broader range of concentrations, especially in the transition regions where α12G shifts from being an activator to an inhibitor. This could help in refining the therapeutic dose range and understanding the concentration-dependent dynamics more accurately.

A: We will further explore the relationship between concentration and hSR’s activator and/or inhibitory function. Selection of dosage must be very careful of clinical use due to it may cause contrary effectiveness of α12G we hope for. We add this point in the discussionin the last part of the discussion. The dose-response assessment of α12G in several NMDA dysfunction animal models does indicate the inverted U-shaped or the part of an inverted U-shaped dose-response but this needs to be confirmed (See Figure 5A, B). We add this point in the limitation part of the discussion.

  1. While the acute pharmacodynamic effects of α12G are promising, long-term studies are necessary to evaluate the potential chronic effects and toxicity. Chronic administration studies in rodent models, assessing both behavioral outcomes and potential nephrotoxicity or hepatotoxicity, would be valuable.

A: We appreciate this comment. Long-term studies and toxicity evaluation will be informative to further develop α12G as a drug candidate. We add this point in the limitation part of the discussion.

  1. The paper would benefit from a comparative analysis of α12G with other known serine racemase modulators, both activators and inhibitors. This comparison could help contextualize α12G's efficacy and safety profile relative to other compounds targeting the same pathway.

A: We were also very curious about the regulatory functions of other known SR modulators. We add this point in the limitation part of the discussion. DMM is the one we investigated (section 3.1). Previous studies are limited by the existing two-step assay for evaluating SR modulators. We found the SR modulator carries concomitant DAAO inhibitory activity. Therefore, previous interpretations of the SR modulator need to be vigorously re-examined (Section 9 of the discussion).

  1. Additional experiments to investigate the downstream effects of α12G on synaptic plasticity, learning, and memory would strengthen the paper. Electrophysiological studies measuring long-term potentiation (LTP) or long-term depression (LTD) in hippocampal slices from treated animals could provide direct evidence of cognitive enhancement.

A: Thank you for your suggestion. Future electrophysiological studies worth pursuing. We add this point in the limitation part of the discussion. We had added a learning memory task (Figure 5C)

  1. The use of open field and prepulse inhibition tests provides initial evidence of α12G's effects on NMDA receptor-mediated behaviors. Expanding the behavioral assessment to include tests for anxiety-like behavior, depression-like symptoms, and other cognitive tasks such as maze learning would provide a more comprehensive profile of α12G's therapeutic potential.

A: We apply a novel object recognition test (NOR) to further investigate the memory-rescue of α12G, please see the results 3.7 and Figure 5C. Other experiments such as a forced swimming test, tail suspension test, and three-chamber sociability test will be pursued. We add this point in the limitation part of the discussion.

  1. The study focuses on intravenous administration of α12G. Exploring alternative administration routes, such as oral or intranasal, could increase the clinical relevance and potential for translational application of the findings. Pharmacokinetic studies comparing different administration routes would be an excellent addition to the paper.

A: We clarified the route of administering α12G which was via oral gavage. We added the section “2.8 Drug Administration”. We agree that “Pharmacokinetic studies comparing different administration routes would be an excellent addition to the paper.” We add this point in the limitation part of the discussion.

Overall, this study represents a significant step forward in the development of modulators for serine racemase as therapeutic agents for CNS disorders. Addressing these suggestions could further solidify the findings and pave the way for translational and clinical research in this promising area.

A: Thanks for your compliment!

Reviewer 2 Report

Comments and Suggestions for Authors

1.       There are several grammatical errors and awkward phrasing throughout the abstract. For example:

"NMDA hyperfunction are involved" should be "NMDA hyperfunction is involved."

"improve the psychotic symptoms and cognition in NMDA hypofunction disorders" could be rephrased to "improve psychotic symptoms and cognition in NMDA hypofunction disorders."

2.       The use of terms like "NMDA-enhancing agents" and "NMDA hypofunction disorders" should be consistent throughout the abstract for clarity.

3.       The description of the study's methodology and results is somewhat unclear. For example:

"to treat C57BL/6 male mice intravenous" could be clarified to "to intravenously treat C57BL/6 male mice."

The description of the outcome of the DMM administration is vague. It would be helpful to specify what the "unexpected outcome" was.

4.       The abstract assumes a high level of prior knowledge from the reader. Explaining key terms and concepts, such as "serine racemase" and "racemization reaction," would make the abstract more accessible to a broader audience.

5.       The abstract should clearly state the main findings of the study and their significance in the context of CNS hypofunction disorders. For example, the sentence "Moreover, α12G treatment can improve major NMDA hypofunction, in animal models of positive symptom and cognitive impairment" could be rephrased to explicitly state how α12G treatment improves these conditions.

6.       The introduction is quite long and covers a wide range of topics. It might be more effective to focus on the key aspects relevant to the study's main objectives and hypotheses.

7.       Some terms and concepts are introduced without sufficient explanation, which could be confusing for readers not familiar with the field. For example, the dual-base mechanism of serine racemase and the distinction between intrasynaptic and extrasynaptic NMDARs could be explained more clearly.

8.       The introduction could be better organized to guide the reader through the background information, the significance of the study, and the research objectives. Using subheadings or paragraphs to separate different topics might help.

9.       Ensure consistency in the use of terms throughout the introduction. For example, "NMDAR" and "NMDA receptor" are used interchangeably; it would be better to stick to one term for clarity.

10.   The citation style seems inconsistent, with some references given in parentheses and others not. It would be best to follow a consistent citation style throughout the introduction.

11.   There are some grammatical errors and awkward phrasings that could be revised for clarity. For example:

12.   "Serine racemase (SR) is an enzyme regulates D-serine" should be "Serine racemase (SR) is an enzyme that regulates D-serine."

13.   "For NMDARS, glycine is located close to intrasynaptic NMDARs, while D-serine is close to the extrasynaptic NMDAR" could be rephrased for clarity.

14.   The introduction should clearly state the main objectives of the study and how it aims to address the gaps in current knowledge. This could be more explicitly stated towards the end of the introduction.

15.   Ensure consistent use of units and abbreviations throughout the section. For example, "mins" should be consistently written as "minutes," and "µl" should be written as "µL" for microliters.

16.   Some procedures are described in a way that might be confusing or lacking in detail. For example:

17.   In section 2.1, the sentence "The cerebral cortex and plasma (n=3-9) were collected, homogenized on ice with 1000 ng/ml tolbutamide (internal standard, IS) in methanol and analyzed by LC-MS/MS..." could be clarified to specify how the samples were prepared and analyzed.

18.   In section 2.2, the description of the PCR and cloning procedures could be made more clear and detailed to ensure reproducibility.

19.   Ensure consistency in formatting throughout the section, such as using the same font size, style, and spacing between paragraphs and sections.

20.   Abbreviations should be defined at their first use in the text. For example, "LC-MS/MS" should be spelled out as "liquid chromatography-tandem mass spectrometry (LC-MS/MS)" when first mentioned.

21.   Use subheadings to clearly separate different experimental procedures and make the section easier to navigate.

22.   In section 2.1, there is a reference to "Marshall Biomedicines 2024, 12, x FOR PEER REVIEW 3 of 14," which seems to be an error and should be removed or corrected.

23.   The section on animal care (2.6) should include more details about the care and handling of the animals, as well as the specific ethical approval number or committee that approved the study.

24.    In sections like 2.7 and 2.8, the experimental design and group assignments could be described more clearly to ensure that readers can understand and potentially replicate the study.

25.   Ensure consistent use of units and abbreviations throughout the section. For example, "mins" should be consistently written as "minutes," and "µM" should be consistently used for micromolar.

26.   Clarity in presenting results: Some results are presented in a way that might be confusing or lacking in detail. For example:

27.   In section 3.1, the sentence "DMM significantly raised D-serine levels in cerebral cortex as well as plasma presumably due to promoting SR racemization" could be clarified to specify how much the levels were raised and provide statistical significance.

28.   In section 3.3, the sentence "The catalytic efficiency of dehydration of L-serine is dramatically enhanced by 2 mM ATP..." could be clarified to specify the conditions under which this enhancement occurs.

29.   Ensure consistency in the use of terms throughout the section. For example, "racemization reaction" and "racemization" are used interchangeably; it would be better to stick to one term for clarity.

30.   References to figures (e.g., Figure 1, Figure 2) should be clearly linked to the corresponding figures, and the figures should be presented in a way that is easy to understand and interpret.

31.   The results should include statistical analysis to support the findings. For example, in section 3.7, the sentence "At the dose of 1 mg/kg, α12G can significantly alleviate PPI deficits at PPI 6 and PPI 10 (p<0.05)" provides a p-value, but it would be helpful to include the statistical test used and the sample size.

32.   There are some grammatical errors and awkward phrasings that could be revised for clarity. For example, "Pharmacodynamic Effectss" in section 3.7 should be "Pharmacodynamic Effects."

33.   Ensure consistency in the use of terms throughout the section. For example, "NMDA function," "NMDA-enhancing agents," and "NMDA hypofunction" are used interchangeably; it would be better to stick to one term for clarity.

34.    Some interpretations of the results are unclear or not well-supported by the data presented. For example:

The statement "We first observed the administration of DMM, pro-drug of malonate, supposed to inhibit SR, paradoxically elevated D-serine levels..." could be clarified to explain why this result is paradoxical and how it relates to the overall findings.

35.   The interpretation of the dose-effect relationship as an "inverted U-shape" should be supported by clear evidence from the results.

36.   "The catalytic efficiency of dehydration of L-serine is higher with a kcat/KM of 253 s-1 M-1 than D-serine dehydration and L-serine racemization, witch with a kcat/KM of 2.4 s-1 M-1 and 17.5 s-1 M-1 respectively..." could be rephrased for clarity.

37.   Ensure consistent use of units and abbreviations throughout the section. For example, "µM" should be consistently used for micromolar, and "kcat/KM" should be consistently formatted.

38.   The discussion should include a section on the limitations of the study and how they might impact the interpretation of the results.

Comments on the Quality of English Language

major editing required

Author Response

  1. There are several grammatical errors and awkward phrasing throughout the abstract. For example:

"NMDA hyperfunction are involved" should be "NMDA hyperfunction is involved."

"improve the psychotic symptoms and cognition in NMDA hypofunction disorders" could be rephrased to "improve psychotic symptoms and cognition in NMDA hypofunction disorders."

A: We have corrected grammatical errors throughout the abstract.

  1. The use of terms like "NMDA-enhancing agents" and "NMDA hypofunction disorders" should be consistent throughout the abstract for clarity.

A: All terms of "NMDA-enhancing agents" and "NMDA hypofunction disorders" now are consistent throughout the article.

  1. The description of the study's methodology and results is somewhat unclear. For example:

"to treat C57BL/6 male mice intravenous" could be clarified to "to intravenously treat C57BL/6 male mice."

The description of the outcome of the DMM administration is vague. It would be helpful to specify what the "unexpected outcome" was.

A: First, the term “intravenous” is replaced by “intravenously” and its abbreviation “IV”. Second, we rephrase the description of the outcome of the DMM administration in the abstract. More details are added in the last paragraph of the introduction.

  1. The abstract assumes a high level of prior knowledge from the reader. Explaining key terms and concepts, such as "serine racemase" and "racemization reaction," would make the abstract more accessible to a broader audience.

A: We rephrase the abstract to explain key terms such as “Serine racemase (SR), the enzyme regulating D-serine levels through both racemization (catalyze from L-serine to D-serine) and b-elimination (degradation of D-serine), emerges as a promising target for bi-directional regulation of NMDA function”. We hope the revision could be more accessible to a broader audience.

  1. The abstract should clearly state the main findings of the study and their significance in the context of CNS hypofunction disorders. For example, the sentence "Moreover, α12G treatment can improve major NMDA hypofunction, in animal models of positive symptom and cognitive impairment" could be rephrased to explicitly state how α12G treatment improves these conditions.

A: We rephrase this part as “Moreover, α12G treatment can improve major NMDA hypofunction disorders including hyperactivity, prepulse inhibition deficit, and memory impairment in animal models of positive symptom and cognitive impairment”.

  1. The introduction is quite long and covers a wide range of topics. It might be more effective to focus on the key aspects relevant to the study's main objectives and hypotheses.

A: We revised the introduction to include necessary background knowledge and the rationale for this study.

  1. Some terms and concepts are introduced without sufficient explanation, which could be confusing for readers not familiar with the field. For example, the dual-base mechanism of serine racemase and the distinction between intrasynaptic and extrasynaptic NMDARs could be explained more clearly.

A: We rephrase this part as “For NMDA receptors, glycine is located close to extrasynaptic NMDA receptors, while D-serine is located close to the synaptic NMDA receptors. It is the consequence of a preferential affinity of synaptic NMDA receptors for D-serine and extrasynaptic NMDA receptors for glycine”. A cell paper “Synaptic and extrasynaptic NMDA receptors are gated by different endogenous coagonists” by Papouin et al. is referenced.

  1. The introduction could be better organized to guide the reader through the background information, the significance of the study, and the research objectives. Using subheadings or paragraphs to separate different topics might help.

A: We add the subheadings to separate different topics in the Introduction.

  1. Ensure consistency in the use of terms throughout the introduction. For example, "NMDAR" and "NMDA receptor" are used interchangeably; it would be better to stick to one term for clarity.

A: "NMDA receptor" is consistent throughout the whole article now.

  1. The citation style seems inconsistent, with some references given in parentheses and others not. It would be best to follow a consistent citation style throughout the introduction.

A: The citation style of references in the introduction is consistent now.

  1. There are some grammatical errors and awkward phrasings that could be revised for clarity. For example:
  2. "Serine racemase (SR) is an enzyme that regulates D-serine" should be "Serine racemase (SR) is an enzyme that regulates D-serine."

A: The error is corrected.

  1. "For NMDARS, glycine is located close to intrasynaptic NMDARs, while D-serine is close to the extrasynaptic NMDAR" could be rephrased for clarity.

A: We have rephrased this part as mentioned in “7.”

  1. The introduction should clearly state the main objectives of the study and how it aims to address the gaps in current knowledge. This could be more explicitly stated towards the end of the introduction.

A: We rephrase section 1.3 “Research objectives.”

  1. Ensure consistent use of units and abbreviations throughout the section. For example, "mins" should be consistently written as "minutes," and "µl" should be written as "µL" for microliters.

A: “Mins” now is written as "minutes" throughout the whole article. "µl" now is written as "µL" throughout the whole article.

  1. Some procedures are described in a way that might be confusing or lacking in detail. For example:
  2. In section 2.1, the sentence "The cerebral cortex and plasma (n=3-9) were collected, homogenized on ice with 1000 ng/ml tolbutamide (internal standard, IS) in methanol and analyzed by LC-MS/MS..." could be clarified to specify how the samples were prepared and analyzed.

A: We rephrase section 2.1 and add more details to clarify how the samples were prepared and analyzed.

  1. In section 2.2, the description of the PCR and cloning procedures could be made more clear and detailed to ensure reproducibility.

A: We add more details of the PCR and cloning procedures. The figure of the PCR and cloning procedures will be presented in supplementary data.

  1. Ensure consistency in formatting throughout the section, such as using the same font size, style, and spacing between paragraphs and sections.

A: We have checked the formatting and make sure it is consistent throughout the section.

  1. Abbreviations should be defined at their first use in the text. For example, "LC-MS/MS" should be spelled out as "liquid chromatography-tandem mass spectrometry (LC-MS/MS)" when first mentioned.

A: It has been corrected.

  1. Use subheadings to clearly separate different experimental procedures and make the section easier to navigate.

A: It has been clearly separated in M&M.

  1. In section 2.1, there is a reference to "Marshall Biomedicines 2024, 12, x FOR PEER REVIEW 3 of 14," which seems to be an error and should be removed or corrected.

A: We didn’t see this reference in section 2.1.

  1. The section on animal care (2.6) should include more details about the care and handling of the animals, as well as the specific ethical approval number or committee that approved the study.

A: We add detailed information on IACUC and the approval number of this study.

  1.  In sections like 2.7 and 2.8, the experimental design and group assignments could be described more clearly to ensure that readers can understand and potentially replicate the study.

A: We add drug administration and group assignments in each behavioral test in section 2.8.

  1. Ensure consistent use of units and abbreviations throughout the section. For example, "mins" should be consistently written as "minutes," and "µM" should be consistently used for micromolar.

A: "mins" now is written as "minutes" throughout the whole article  

"µl" now is written as "µL" throughout the whole article.

  1. Clarity in presenting results: Some results are presented in a way that might be confusing or lacking in detail. For example:
  2. In section 3.1, the sentence "DMM significantly raised D-serine levels in the cerebral cortex as well as plasma presumably due to promoting SR racemization" could be clarified to specify how much the levels were raised and provide statistical significance.

A: We add the annotations to clarify how much the levels were raised and provide statistical significance (Section 3.1).

  1. In section 3.3, the sentence "The catalytic efficiency of dehydration of L-serine is dramatically enhanced by 2 mM ATP..." could be clarified to specify the conditions under which this enhancement occurs.

A: The sentence now is rephrased as “The catalytic efficiency of L-serine dehydration is dramatically enhanced by adding 2 mM ATP in the presence of 200 mM L-serine at 37 °C, which leads to a 31-fold increase in enzyme activation, with a change of kcat/KM from 8.1 to 253 s-1 M-1.”

  1. Ensure consistency in the use of terms throughout the section. For example, "racemization reaction" and "racemization" are used interchangeably; it would be better to stick to one term for clarity.

A: Now, the term " racemization " is consistent throughout the whole article.

  1. References to figures (e.g., Figure 1, Figure 2) should be clearly linked to the corresponding figures, and the figures should be presented in a way that is easy to understand and interpret.

A: All figures are clearly referenced. Several figures are reformatted to improve the qualities (e.g., Figure 2, Figure 3B, Figure 4A, and Figure 4B)

  1. The results should include statistical analysis to support the findings. For example, in section 3.7, the sentence "At the dose of 1 mg/kg, α12G can significantly alleviate PPI deficits at PPI 6 and PPI 10 (p<0.05)" provides a p-value, but it would be helpful to include the statistical test used and the sample size.

A: The statistical tests, p values, and sample numbers are provided in the M&M and results sections. 

  1. There are some grammatical errors and awkward phrasings that could be revised for clarity. For example, "Pharmacodynamic Effectss" in section 3.7 should be "Pharmacodynamic Effects."

A: This error has been corrected.

  1. Ensure consistency in the use of terms throughout the section. For example, "NMDA function," "NMDA-enhancing agents," and "NMDA hypofunction" are used interchangeably; it would be better to stick to one term for clarity.

A: As we mentioned in question 2, all terms of "NMDA-enhancing agents" and "NMDA hypofunction disorders" now are consistent throughout the article. However, we suppose the “NMDA function” is a different concept that could not be replaced with the other two terms.

  1.  Some interpretations of the results are unclear or not well-supported by the data presented. For example:

The statement “We first observed the administration of DMM, pro-drug of malonate, supposed to inhibit SR, paradoxically elevated D-serine levels”…" could be clarified to explain why this result is paradoxical and how it relates to the overall findings.

A: Thank you for your suggestion. We rephrase this sentence as “We observed the IV administration of DMM, pro-drug of malonate, supposed to inhibit SR in vitro (Vorlová et al., 2014; Rani et al., 2020), in fact, elevated D-serine levels in the cerebral cortex and plasma of C57BL/6 mice.” The references stated malonate was considered an SR inhibitor and downregulated D-serine in in vitro studies. But we found out it elevated D-serine otherwise.

  1. The interpretation of the dose-effect relationship as an "inverted U-shape" should be supported by clear evidence from the results.

A: According to Figure 3B, the concentration-dependent response curve for the racemization reaction (% on the Y axis, drug concentration on the X axis) exhibits an inverted U-shape.

  1. "The catalytic efficiency of dehydration of L-serine is higher with a kcat/KM of 253 s-1 M-1 than D-serine dehydration and L-serine racemization, which with a kcat/KM of 2.4 s-1 M-1 and 17.5 s-1 M-1 respectively..." could be rephrased for clarity.

A: We rephrase this sentence as “The catalytic efficiency of dehydration of L-serine is higher with a kcat/KM of 253 s-1 M-1 than D-serine dehydration (kcat/KM=2.4 s-1 M-1) and L-serine racemization (kcat/KM=17.5 s-1 M-1), in the presence of 2 mM ATP.”

  1. Ensure consistent use of units and abbreviations throughout the section. For example, "µM" should be consistently used for micromolar, and "kcat/KM" should be consistently formatted.

A: These abbreviations now are consistently formatted.

  1. The discussion should include a section on the limitations of the study and how they might impact the interpretation of the results.

A: We added a section discussing the limitations of our study in the last part of the discussion.

Round 2

Reviewer 1 Report

Comments and Suggestions for Authors

I have no comments

Reviewer 2 Report

Comments and Suggestions for Authors

All the comments are addressed satisfactorily